# Efficient self-assembly of heterometallic triangular necklace with strong antibacterial activity

Gui-Yuan Wu[1], Xueliang Shi [1✉], Hoa Phan[2], Hang Qu[3], Yi-Xiong Hu[1], Guang-Qiang Yin[1], Xiao-Li Zhao[1], Xiaopeng Li [4], Lin Xu[1], Qilin Yu [5✉] & Hai-Bo Yang [1✉]

Sophisticated mechanically interlocked molecules (MIMs) with interesting structures, properties and applications have attracted great interest in the field of supramolecular chemistry. We herein report a highly efficient self-assembly of heterometallic triangular necklace **1** containing Cu and Pt metals with strong antibacterial activity. Single-crystal X-ray analysis shows that the finely arranged triangular necklace **1** has two racemic enantiomers in its solid state with intriguing packing motif. The superior antibacterial activity of necklace **1** against both standard and clinically drug-resistant pathogens implies that the presence of Cu (I) center and platinum(II) significantly enhance the bacterium-binding/damaging activity, which is mainly attributed to the highly positively charged nature, the possible synergistic effect of heterometals in the necklace, and the improved stability in culture media. This work clearly discloses the structure-property relationships that the existence of two different metal centers not only facilitates successful construction of heterometallic triangular necklace but also endows it with superior nuclease properties and antibacterial activities.

---

[1] Shanghai Key Laboratory of Green Chemistry and Chemical Processes, School of Chemistry and Molecular Engineering, East China Normal University, 3663N. Zhongshan Road, Shanghai 200062, P. R. China. [2] Vinh University, 182 LeDuan Street, Vinh, Vietnam. [3] State Key Laboratory of Physical Chemistry of Solid Surfaces, Collaborative Innovation Center of Chemistry for Energy Materials (iChEM) and College of Chemistry and Chemical Engineering, Xiamen University, Xiamen 361005, China. [4] Department of Chemistry, University of South Florida, Tampa, FL 33620, USA. [5] Key Laboratory of Molecular Microbiology and Technology, Ministry of Education, College of Life Sciences, Nankai University, Tianjin 300071, P. R. China. ✉email: xlshi@chem.ecnu.edu.cn; yuqilin@mail. nankai.edu.cn; hbyang@chem.ecnu.edu.cn

Mechanically-interlocked molecules (MIMs) such as catenanes[1–6], rotaxanes[7–18], molecular necklaces[19,20], and molecular knots[21,22] have been a main research focus in supramolecular chemistry due to their innately topologically nontrivial architectures, thus always challenging the imagination and skills of synthetic chemists[23–27]. Molecular necklaces, as important members in MIMs family, are derived from catenanes, in which three or more side rings as molecular "beads" are threaded onto a central ring as molecular "chains"[28]. The early research of molecular necklace could be traced back to the work reported by Sauvage et al., who generated a mixture of molecular necklaces by accident in a very low yield and characterized their structures by using electrospray ionization mass spectrometry (ESI-MS)[29]. Moreover, Stoddart et al.[30–32], Chiu et al.[33], Grubbs et al.[34], and Wu et al.[35] also reported a series of molecular necklaces through covalent synthesis assisted by weak donor–acceptor π–π interactions or dynamic covalent chemistry, but the yield was also relatively low. Notably, Kim and coworkers realized the highly efficient assembly of molecular necklaces by taking advantage of coordination bonds and cucurbituril-based host–guest chemistry, which is definitely a big breakthrough[36–38]. Recently, Li, Stang, and co-workers have presented the highly efficient construction of the largest molecular necklace so far through hierarchical self-assembly involving coordination interactions and the subsequent host–guest chemistry[39,40].

Generally, a well-established molecular necklace is self-assembled from three main components: molecular axis as chain, macrocycle as bead and molecular linker that joints them together (Fig. 1). The prevalent strategy to design molecular necklaces is mainly based on threading and ring-closing two processes, basically driven by noncovalent interactions such as host–guest interactions, coordination bonds, dynamic covalent chemistry, etc. Obviously, metals, as the "gems" of molecular necklaces, are of great importance to the efficient synthesis of metallic necklaces[41]. In particular, metals in molecular necklace would not only facilitate the threading process due to their template effect but also participate in the ring-closing process via coordination-driven self-assembly[42,43]. Moreover, the existence of metals will bring some interesting properties and applications of the resultant necklace. However, most of the documented metallic necklaces only involve one kind metal in their architectures, and no heterometallic molecular necklace has been reported so far. Therefore, there is a high demand for the design of functionalized molecular necklaces that possess heterometal centers and intriguing properties for applications.

It's worth-noting that, though a major breakthrough in molecular necklaces regarding to their impressive structures as well as well-developed synthetic strategy has been achieved, the study of their properties and potential applications have persistently lagged behind expectations. To the best of our knowledge, the focus of the chemistry of MIMs, especially for molecular necklaces, has still stayed on molecular design and synthesis stage. Studies involving their applications are hardly seen at present in this area. The intrinsic nature of coordination-driven self-assemblies including their high dynamic metal–ligand coordination bonds, diverse metal ions, positively charged molecular nature, together with their highly tunable coordination geometries endows them with great merits in biomedical applications[44]. Indeed, the biomedical applications of coordination-driven self-assembly of metallacycles and metallacages have been systematically investigated in last decade, and they can serve well as anticancer agents, drug delivery systems, biosensors, DNA intercalators, antibacterial agents and so on[45,46]. It is anticipated that molecular necklaces, the distinct type of metal coordination-driven self-assemblies, may also be developed for biomedical applications, especially when their biomolecule-interaction efficiency and stability are highly improved.

The emerging drug-resistant bacterial pathogens have been becoming great threats to human health, and thus provoke the development of advanced antibacterial agents[47,48]. The bacterial DNA, cell wall, and plasma membrane are critical components of bacterial cells essential for survival and growth[49]. The negatively charged property of DNA, bacterial cell wall (displaying negatively charged lipopolysaccharides or teichoic acid), and plasma membrane (exposing negatively charged phospholipids)[50] indicates that they are realistic targets of the positively charged metal coordinating complexes. The coordination-driven self-assembly of the positively charged metallosupramolecular complexes just provides a powerful platform to enhance the electric charges for severely disruption of the targeted DNA/cell wall/plasma membrane, and to improve the stability for prolonging the interaction time between the necklaces and the bacterial cells. Inspired by the recent development of biomedical and biochemical applications of coordination-driven self-assembly, we envision that molecular necklaces possessing multiple metals as well as the enhanced electric charges and stability may potentially serve as efficient DNA intercalators and bacterial cell wall/plasma membrane-disrupting agents, thus opening a promising window to their biomedical applications.

In this study, we report the construction of a heterometallic triangular necklace **1** containing both Cu and Pt metals with

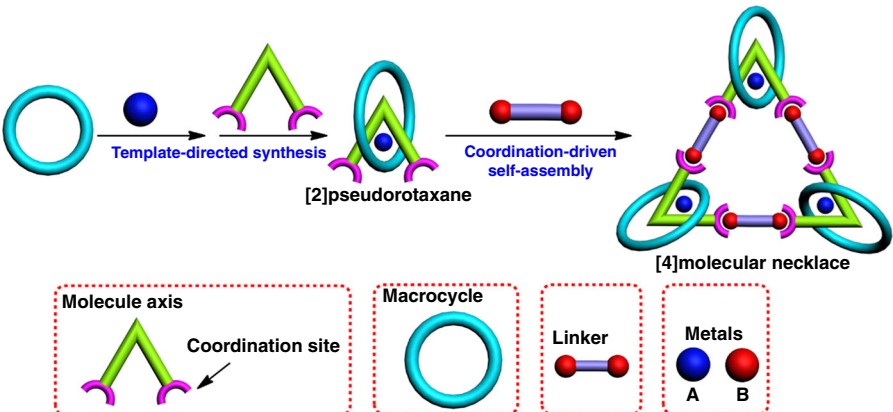

**Fig. 1 The concept of molecular necklace design.** The cartoon presents the "threading-followed-by-ring-closing" approach for construction of heterometallic triangular necklace.

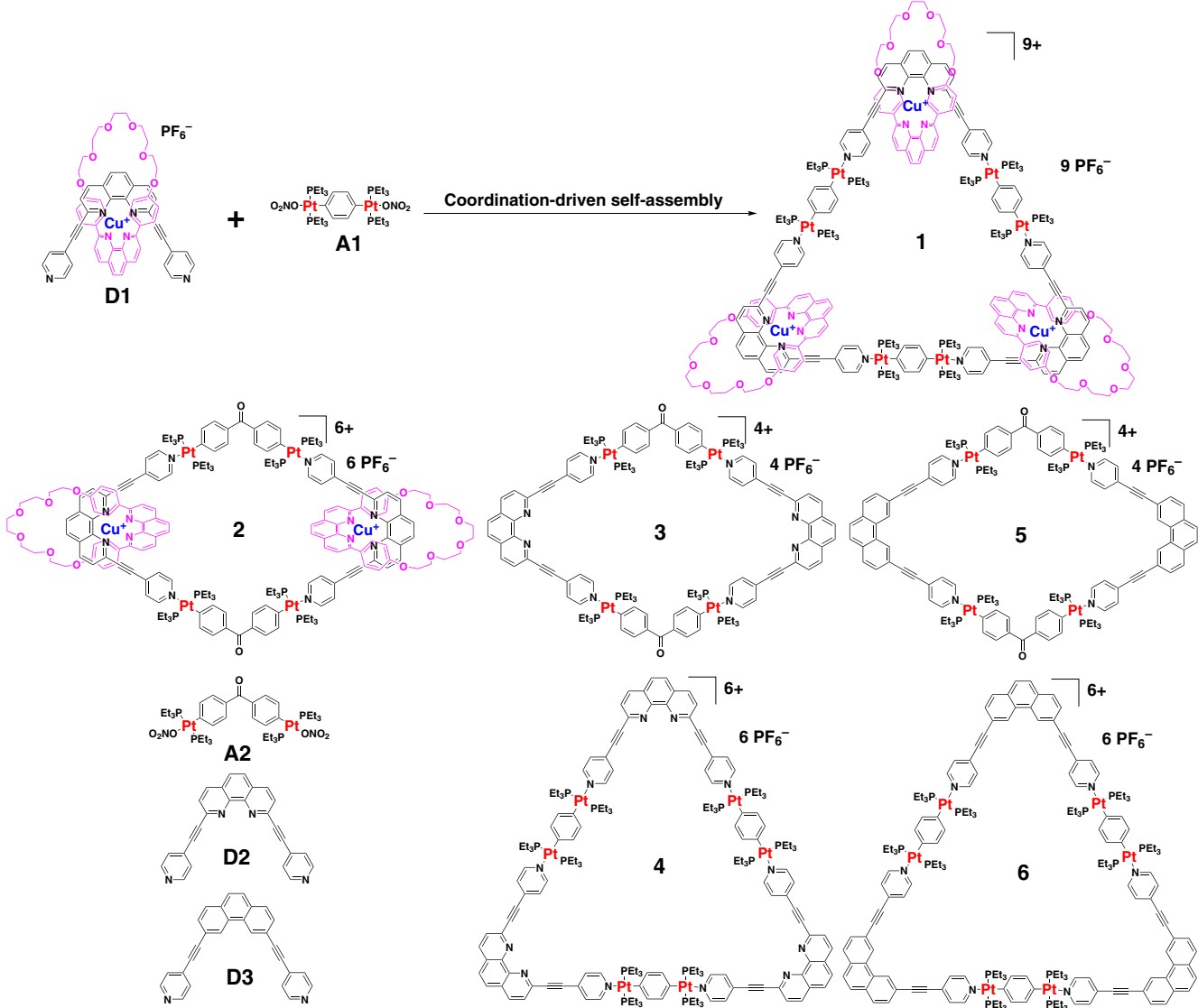

**Fig. 2 Synthetic route and chemical structures of necklace 1 and related molecules used in this study.** The self-assembly route of necklace **1** was demonstrated in the front image.

strong antibacterial activity through a highly efficient "threading-followed-by-ring-closing" approach driven by metal ligand coordination (Fig. 1). The elegant structure of necklace **1** is successfully determined by X-ray crystallographic analysis, revealing that the finely arranged triangular necklace **1** has two racemic enantiomers in its solid state with intriguing packing motif. The existence of two different metal centers not only facilitates the successful construction of necklace **1** but also endows it with superior nuclease properties and activities. Our studies further show that the excellent antibacterial activity might be mainly attributed to the synergistic effect of heterometals in the necklace, which enhances its bacterium-binding and cell wall/plasma membrane-disrupting capacity for killing the bacterial cells.

## Results

**Synthesis**. The major challenge for the synthesis of heterometallic molecular necklace in this study is the compatibility of the selected two coordination motifs. Therefore, two orthogonal 2,9-disubstituted Cu(I)-bis(phenanthroline)s ([Cu(phen)$_2$]$^+$) and

platinum(II)–pyridine coordination motifs are particularly selected as they are mutually compatible, i.e., 2,9-disubstituted [Cu(phen)$_2$]$^+$ complex will not affect platinum(II)–pyridine coordination process[51,52], probably duo to the steric hindrance[53]. The synthetic route to molecular necklace **1** is depicted in Fig. 2. Pseudorotaxane **D1** consisting of two functional ligands of pyridine and phenanthroline was synthesized from the starting materials **M1** and **D2** according to a literature method[54] with minor modifications (Supplementary Fig. 1). Then heterometallic triangular necklace **1** was synthesized through the straightforward coordination-driven self-assembly between pyridine donor **D1** and diplatinum (II) acceptor unit **A1** in nearly quantitative yield. Notably, pseudorotaxane **D1** could generated in situ without the further purification. Therefore, the heterometallic triangular necklace **1** was achieved in quantitative one-pot synthesis simply by mixing **M1** and **D2** with **A1** (Supplementary Fig. 2). More impressively, the necklace **1** could be isolated in pure form, which enables us to further study its properties and potential applications. In order to elaborate the challenge of the synthesis of heterometallic triangular necklace, some additional controlled

experiments were carried out. For instance, another molecular necklace **7** through coordination-driven self-assembly between 3,8-disubstituted $[Cu(phen)_2]^+$ **D5** and the corresponding diplatinum (II) acceptors **A3** was proposed (Supplementary Figs. 8 and 9). However, the complicated $^1H$ and $^{31}P$ nuclear magnetic resonance (NMR) spectrum indicated the unsuccessful self-assembly process (Supplementary Fig. 20). This indicates that not all the ligands are applicable to self-assemble to form the designed necklace, thus proving the challenge to synthesize such heterometallic molecular necklace. It indicates that 3,8-disubstituted $[Cu(phen)_2]^+$ and platinum(II)–pyridine these two coordination motifs are not compatible, i.e., the controlled experiment demonstrated that 3,8-disubstituted $[Cu(phen)_2]^+$ is likely to interfere the platinum(II)–pyridine coordination process because of the potential competing coordination reaction between platinum(II) and nitrogen in 3,8-disubstituted $[Cu(phen)_2]^+$ (Supplementary Figs. 10 and 20). For comparison, [3]catenane **2** as the analog of **1** with two $[Cu(phen)_2]^+$ units was also synthesized by using the same protocol (Supplementary Fig. 3). In order to systematically investigate the subsequent biological activity of necklace, we further synthesized a series of metalla-cycles featuring only platinum(II)–pyridine coordination moieties without copper metal (i.e., **3**, **4**, **5**, and **6**) for comparison (Supplementary Figs. 4–7, 21–24). **1** and **2** were well characterized with one-dimensional (1-D) NMR, two-dimensional correlation spectroscopy, diffusion-ordered NMR spectroscopy (DOSY), nuclear overhauser effect spectroscopy, and electrospray ionization time-of-flight mass spectrometry (ESI-TOF-MS) (Supplementary Figs. 11–19). The structure of pseudorotaxane **D1** and necklace **1** were also unambiguously confirmed by X-ray crystallographic analysis.

**NMR and mass characterizations**. Multinuclear NMR ($^1H$ and $^{31}P$) spectroscopy was applied to characterize the structural information of the necklace **1** by comparing with the corresponding ligands. The $^{31}P$ NMR spectrum of the necklace **1** displayed a sharp singlet peak at $\delta$ 14.24 ppm, which significantly shifted upfield from the starting platinum acceptor **A1** by approximately 5.27 ppm (Fig. 3a). This change, as well as the decrease of $^{31}P$-$^{195}Pt$ coupling constant ($ca.$ $\Delta J = -212.5$ Hz) is consistent with the π-back donation from the platinum atoms, indicating the successful coordination between **D1** and **A1**. Additionally, the necklace **1** exhibited a well-resolved $^1H$ NMR spectrum (Fig. 3b), where the protons of the pyridine rings exhibited downfield shifts ($H_\alpha$: 8.27–8.75 ppm; $H_\beta$: 6.43–6.88 ppm) resulting from the loss of electron density upon coordination of the pyridine N atom with the Pt(II) metal center. Moreover, the 2D-DOSY analysis of the necklace **1** in acetone-$d_6$ revealed a single band at $\log D \approx -8.8$ (Fig. 3c). Besides, the ESI-MS-TOF mass spectrometry provided further evidence for the formation of the target interlocked assembly. In the mass spectrum of the necklace **1** (Supplementary Fig. 13), peaks at $m/z$ = 1644.80 and 1287.21 were observed, corresponding to $[M-4PF_6]^{4+}$ and $[M-5PF_6]^{5+}$ moieties, respectively, where $M$ represents the intact assemblies. The NMR and mass analysis of [3]catenane **2** experienced a phenomenon similar to necklace **1**. Therefore, the NMR and mass results both preliminarily proved the successful construction of the necklace **1** and [3]catenane **2**, verifying the rationality of the developed synthetic strategy.

**Crystallographic analysis**. Single crystal suitable for X-ray crystallographic analysis was obtained for pseudorotaxane **D1** by slow evaporation of its acetone solution (Fig. 4a). A racemic mixture of enantiomers was found in the single crystal of **D1**, which is attributed to the intrinsically chiral nature of $[Cu(phen)_2]^+$

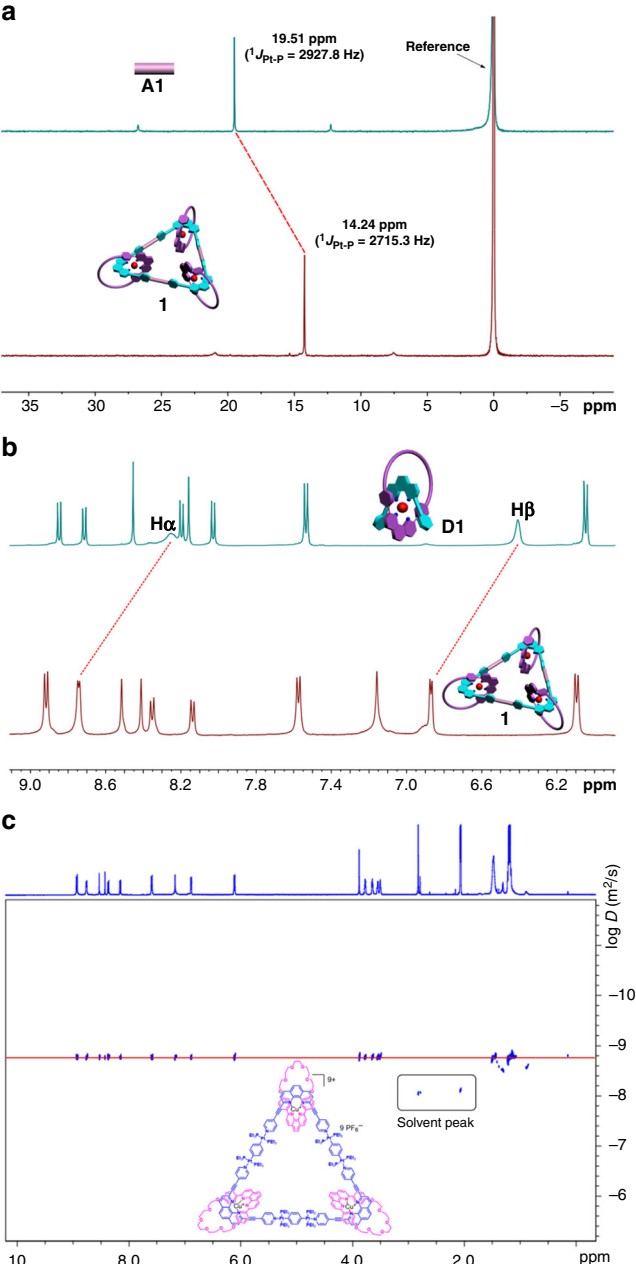

**Fig. 3 NMR spectra of necklace 1 and related molecules. a** Partial $^{31}P$ NMR spectra of **1** and **A1** (202 MHz, acetone-$d_6$, 298 K). **b** Partial $^1H$ NMR spectra of **1** and **D1** (500 MHz, acetone-$d_6$, 298 K). **c** 2-D DOSY spectrum of **1** (500 MHz, acetone-$d_6$, 298 K).

unit[55]. In its packing structure, multiple intramolecular and intermolecular interactions were observed. Specifically, one pyridyl face-to-face packed with intramolecular phenanthroline with a centroid-to-centroid distance of 3.813 Å, while the other pyridyl stacked with phenoxy groups of the neighboring enantiomer with centroid-to-centroid distance of 4.183 Å. Intramolecular π–π stacking between phenoxy group and phenanthroline was also observed with a centroid-to-centroid distance of 3.533 Å. Consequently, two enantiomers alternatively packed together into a column along the $a$ axis. Typically, it is very difficult to obtain good single crystals of large cationic macrocycles based on Pt(II)-N(pyridine) coordinative bonds, not to mention such complicated molecular necklace. Fortunately, single crystal of

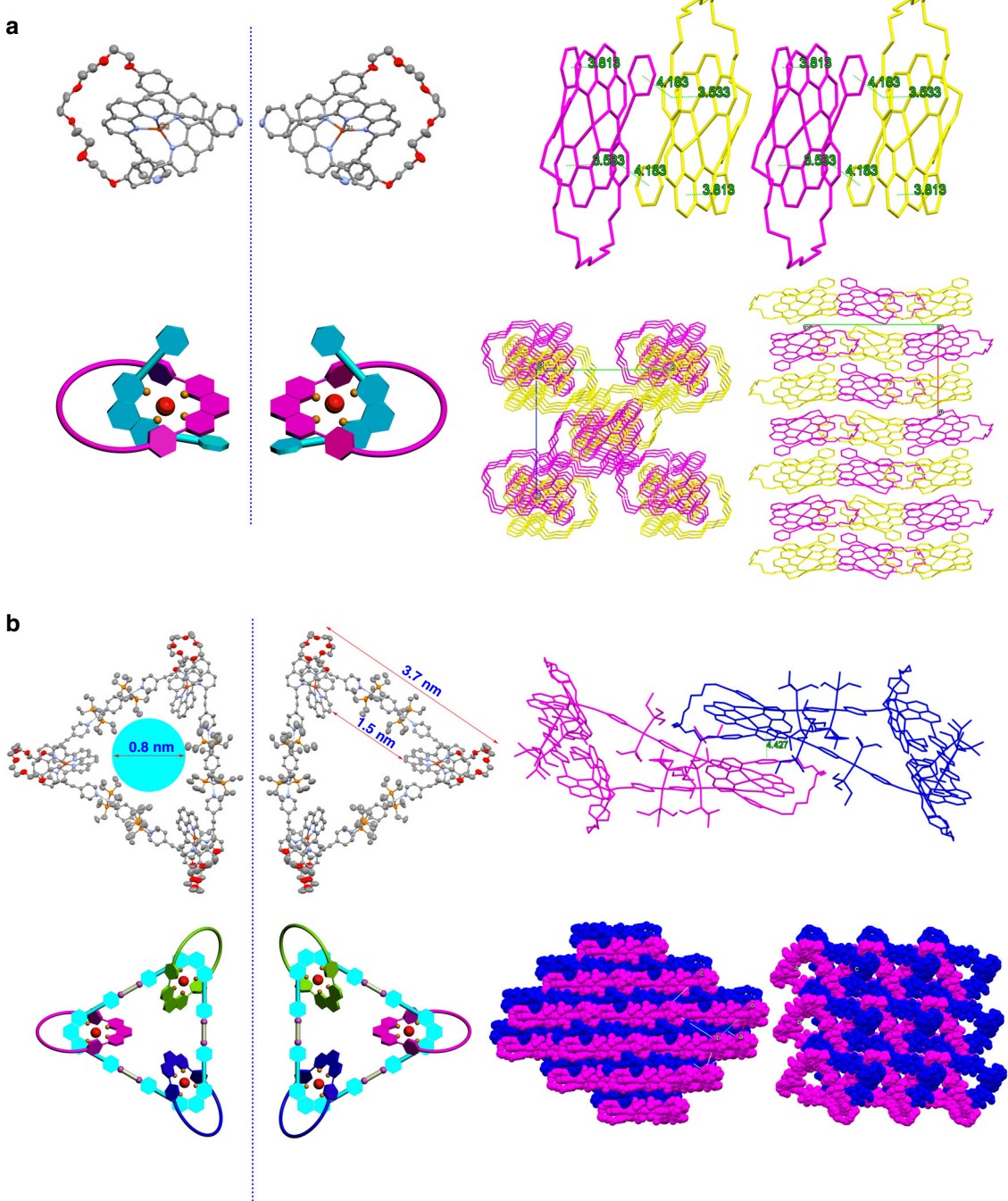

**Fig. 4 Single-crystal X-ray structure analysis.** ORTEP drawings and 3D packing structures (both top-view and side-view) of **a D1** and **b 1**. The two enantiomers of **D1** and **1** are also presented in cartoon format. Hydrogen atoms and $PF_6^-$ anions are omitted for clearance.

heterometallic triangular necklace with good quality was obtained by slow diffusion of ether into acetone solution (Fig. 4b). The compound crystallizes in P-1 space group with the asymmetric unit is the whole triangular necklace **1** molecule. The large void inside the triangular necklace is filled with disordered solvents, which are removed using SQUEEZE routine of the Platon program during the crystal structure refinement[56]. To the best of our knowledge, this is the first time to present a quality crystal of heterometallic molecular necklace based on [3 + 3] Pt(II)-N cationic metallacycle. As shown in Fig. 4b, three polyether phenanthroline macrocycles were ideally threaded onto a [3 + 3] Pt (II)-N cationic metallacycle, thus resulting in a finely arranged

triangular necklace surrounded by nine hexafluorophosphate anions. Two enantiomers were also found in the crystal of **1** on account to the orientation of $[Cu(phen)_2]^+$ unit. The size of the necklace is also very impressive with the exterior length of approximately 37 Å and internal length of approximately 15 Å, leading to a cavity with diameters of ~8 Å. Regarding to the packing motif, three $[Cu(phen)_2]^+$ units exhibited strong π–π stacking with the adjacent molecules with a centroid-to-centroid distance of 4.427 Å. As a result, every triangular molecule in solid state was arranged very closely to each other and stacked in a stagger packing mode, thereinto two enantiomers in each layer was alternatively aligned. The casual discovery of the chirality in

molecular necklace would enrich the recent development of chiral chemistry in MIMs such as rotaxanes and catenanes[57]. The above results also highlight that the stoichiometry and position of Cu (I)-contained donor and the Pt(II)-contained acceptor unit can be precisely integrated into the well-defined molecular necklace in such coordination-driven self-assembly process, which is of high importance to the following biomedical applications.

**DNA-cleaving and antibacterial activities**. DNA, a double helix carrying the genetic instructions, is of high importance for most of organisms. DNA is the primary target for most of anticancer drugs and anti-infection agents[58]. Investigations on DNA cleavage and developments of efficient chemical nucleases have attracted wide interest owing to their potential applications as promising therapeutic agents as well as diagnostic structural probes to analyze DNA information. Among all the chemical nucleases reported so far, transition metal complexes have obvious advantages because metal complexes with natural cationic character particularly favor the redox, hydrolysis and other photoreactions, thus leading to strong affinity to DNA[59,60]. For example, $[Cu(phen)_2]^+$ units have been widely studied as chemical nucleases for highly efficient DNA cleavage since the redox properties of the metal could not only promote the reactive oxygen species (ROS) generation but also have strong non-covalent interactions with DNA[61]. Recent studies have demonstrated that ditopic coordination compound containing copper and platinum centers displayed some intriguing nuclease properties with the enhanced DNA cleavage efficiency[62–65]. However, very few mechanically-interlocked molecules, especially molecular necklaces, have demonstrated their application prospects in this area[66,67]. On this basis, the study of the nuclease properties and application in DNA cleavage of the designed heterometallic molecular necklace **1** is of great interest. The DNA-cleavage of necklace **1** was then first systematically studied and compared with the above designed [3]catenane **2**, metallacycles **3–6**, pseudorotaxane **D1** as well as di-Pt(II) acceptors **A1** and **A2**. Indeed, DNA cleavage assay revealed that both the molecular necklace **1** and [3]catenane **2** had much higher DNA cleavage activity than the donor **D1**, the acceptors **A1** and **A2**, and the metallacycles **3**, **4**, **5** and **6** (Supplementary Fig. 25). Especially, necklace **1** led to thorough DNA degradation in 10 min when their concentrations reached up to 6 μM, indicating the higher cleavage activity than [3]catenane **2** (Supplementary Fig. 26). The DNA cleavage activity of necklace **1** was also observed in the **1**-treated bacterial cells, which showed remarkable intracellular DNA fragmentation as compared to the control cells (Supplementary Fig. 27). These results implied that necklace **1** might have the highest biological activity among the tested molecules. Meanwhile, the Cu(I)-containing molecule **D1**, rather than the Cu(I)-free molecules (i.e., **A1**, **A2**, **3**, **4**, **5**, and **6**), also led to obvious DNA degradation, but its DNA cleavage efficiency was much lower than the necklace (Supplementary Fig. 25). This observation indicated that Cu(I) plays a critical role in DNA cleavage by the necklace, and its efficiency was remarkably enhanced by the Pt(II)-containing acceptor.

Strong DNA cleavage activity of the coordination complexes commonly indicates high antibacterial ability[68–70]. Moreover, recent studies have further unveiled some interesting findings that metallacycles and metallacages exhibited strong cell wall (together with plasma membrane)-intercalating ability and antibacterial activity[44–46,71]. Considering the fact that molecular necklace **1** has combinational positive charges, π–π stacking ability, platinum (II)–pyridine coordination and $[Cu(phen)_2]^+$ units, we speculate that it may possess strong antibacterial activity. The positively charged molecular necklace **1** might possess antibacterial activity

owing to its binding with the negatively charged components of the cell wall/plasma membrane, e.g., liposaccharides (LPS) and phospholipids. The binding ability of the molecular necklace to the bacterial cells was then investigated by both glass-adhering test and dynamic light scattering (DLS) analysis using the well-known bacterial pathogen *Pseudomonas aeruginosa*. Confocal observation of the 4′,6-diamidino-2-phenylindole (DAPI)-stained cells on the glass surfaces revealed that the glass surface coated with the necklace **1** could bind abundant bacterial cells, while the surfaces coated with [3]catenane **2** or the metallacycles **3–6** only had weak bacterial binding ability (Fig. 5a). Fluorescence intensity quantification further indicated that **1** could bind 1-fold more bacterial cells than [3]catenane **2**, and 2–3-fold more cells than the metallacycles without copper metals (Supplementary Fig. 28). DLS analysis of the molecule-bacteria interaction suspension further showed that necklace **1** led to the higher size distribution of the bacterial groups (from 620–1980 nm to 3760–4400 nm) than that of [3]catenane **2** (to 750–3548 nm), indicating much more severe cell aggregation caused by the binding of **1** (Supplementary Fig. 29a). In contrast, the metallacycles **3–6** without copper metals had no remarkable effect on the size distribution of the bacterial groups (Supplementary Fig. 29b), confirming the important role of the molecular necklace on cell binding ability. Moreover, **1**-coated glass could adsorb much higher levels of LPS than other complexes (Fig. 5b). These observations suggested that the heterometallic necklace **1** featured strong bacterial binding ability, which is most likely because the necklace **1** has more positive charges and consequently stronger interaction with the cell wall LPS.

The strong binding of the heterometallic necklace with the bacterial cells might result in severe plasma membrane damage and consequent cell death. We then used the 5(6)-carboxyl fluorescein (CF)-leakage model to evaluate the ability of the molecular necklace to damage the plasma membrane[72–74]. Expectedly, while [3]catenane **2** and the metallacycles **3–6** only led to weak and slow CF release (<30% even after 90 min) from the 1,2-dioleoyl-sn-glycero-3-phosphocholine (DOPC) liposomes, the heterometallic necklace **1** caused drastic CF release (~70% in 15 min, and >80% after 90 min) (Fig. 5c), indicating the strong plasma membrane-damaging ability of **1**.

Owing to the strong bacterium-binding and cell membrane-damaging ability of necklace **1**, we expected that it may severely kill the bacterial cells. Colony forming unit assays showed that, while [3]catenane **2** and the metallacycles **3–6**, together with their donor **D1**, the Cu$^+$ precursor Cu(MeCN)$_4$PF$_6$, and the acceptors (**A1** and **A2**), only caused a decrease in cell viability to 30–65%, the heterometallic necklace **1** could kill almost all of the bacterial cells (Fig. 5d, e, Supplementary Fig. 30). Statistical analysis also revealed that necklace **1** had much lower IC$_{50}$ against the bacterial cells than that of the other complexes (Supplementary Table 1). More importantly, heterometallic necklace **1**, as compared to the control complexes, also showed excellent antibacterial activity to the clinically isolated drug-resistant pathogens, such as the ciprofloxacin/penicillin double-resistant *P. aeruginosa* strain (IC$_{50}$ = 3.98 μM), the penicillin/tetracycline *Escherichia coli* strain (IC$_{50}$ = 1.84 μM), and the multidrug-resistant *Staphylococcus aureus* (MRSA) strain (IC$_{50}$ = 5.16 μM), indicating that it may be a superior candidate of antibacterial agents for fighting against multidrug-resistant pathogens.

An interesting observation is that all of the Cu(I)-containing molecules (i.e., **1**, **2**, **D1**, and Cu(MeCN)$_4$PF$_6$) had higher antibacterial activity than the Cu(I)-free molecules (i.e., **3**, **4**, **5**, **6**, **A1**, and **A2**) (Fig. 5d, Supplementary Fig. 30, Table S1). In addition, we also found that all of the Cu(I)-containing molecules have remarkably stronger LPS-binding and membrane-damaging ability than the other molecules (Supplementary Fig. 31). These

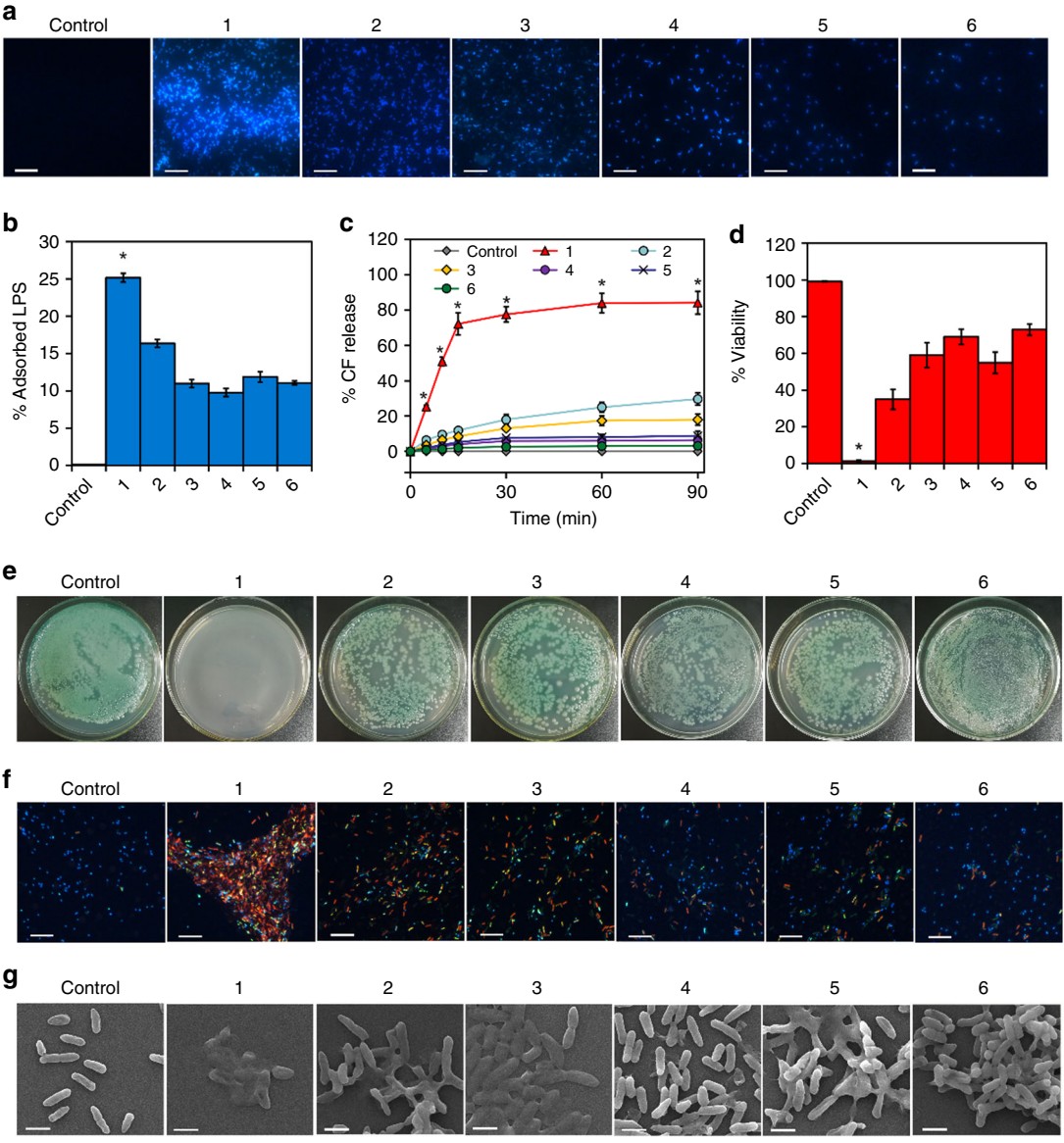

**Fig. 5 Binding-related antibacterial activity of the six designed metallacycles. a** Bacterial adsorption of the *P. aeruginosa* cells (stained by 4',6-diamidino-2-phenylindole (DAPI)) by the metallacycles on glass surfaces (Scale bar = 10 μm). **b** Adsorption activity of the metallacycles with the bacterial liposaccharides. **c** Liposome damage activity of the metallacycles. The released fluorescence dye CF from the liposomes indicates damage of the liposomes caused by the metallacycles. **d** Bacterial viability revealed by CFU assays with the molecular concentration of 8 μM. **e** Growth of the treated *P. aeruginosa* cells on the Luria–Bertani (LB) medium plates. **f** Confocal images of the DAPI-propidium iodide (PI) double-stained *P. aeruginosa* cells (Scale bar = 10 μm). The PI-positive (red) cells indicated the dead cells. Noting that the molecular necklace 1 strongly bound the bacterial cells and induced bacterial aggregation, leading to severe cell death. **g** SEM images of the treated cells (Scale bar = 1 μm). Noting that the molecular necklace 1 strongly bound the bacterial cells and induced cell collapse. Data are represented as mean ± SD (*n* = 3). Asterisk (*) indicates significant difference between the molecular necklace 1 group and the other groups (*P* < 0.05).

results indicated that Cu(I) plays an important role in killing bacterial cells by enhancing the interaction between **1** and the cell membrane components. Notably, only the cooperation of the Cu (I)-contained donor and the Pt(II)-contained acceptor in the necklace **1** generated the highest cell wall/plasma membrane-binding/damaging and bacterial killing ability since the Cu(I)-free matallacycle **4** and the Pt(II)-free donor **D1** exhibited much lower antibacterial capacity than necklace **1** (Fig. 5, Supplementary Fig. 30).

Coordination-driven formation of the necklace **1** framework is also critical for efficiently killing the pathogens. To prove this contribution, a control experiment was carried out. The

antibacterial activities of the ligand mixtures of **D1** + **A1′** or **D1** + **A2′** have been surveyed. Notably, the physical blends of **D1** + **A1′** or **D1** + **A2′** could not self-assemble into the necklace since Pt was protected by iodide or bromide moieties. It was found that, compared to the necklace **1**, the mixtures of **D1** + **A1′** or **D1** + **A2′** exhibited much weaker bacterium-killing efficiency (Supplementary Fig. 32). These results further implied the significance of necklace formation in bacterial killing. From the above discussion, we may conclude that stoichiometric Cu(I)-contained donor and the Pt(II)-contained acceptor can be precisely integrated into the platform of necklace **1**, which is highly conducive to its outstanding bacterial killing ability.

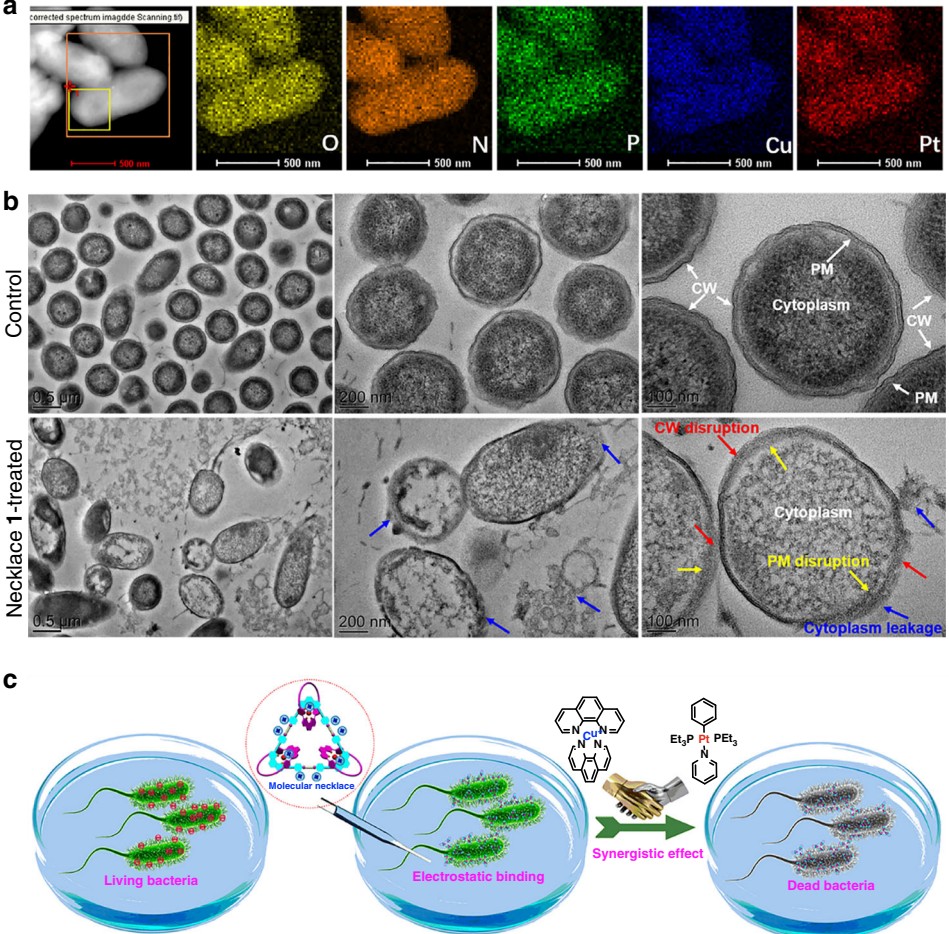

**Fig. 6 Contribution of bacterial binding and cell wall/plasma membrane disruption to bactericidal activity of necklace 1. a** EDS mapping of the main elements (i.e., O, N, P, Cu, and Pt) on the bacterial cell surface (scale bar = 500 nm). **b** TEM observation of ultrathin sections of the control and necklace 1-treated cells (after 30 min of treatment). **i** and **iv** Scale bar = 500 nm. **ii** and **v** Scale bar = 200 nm. **iii** and **vi** Scale bar = 100 nm. The red, yellow, and blue arrows indicate cell wall (CW) disruption, plasma membrane (PM) disruption, and cytoplasm leakage, respectively. Ultrathin sections were cut with a diamond knife, collected on 200-mesh copper grids, stained with uranyl acetate and lead citrate, and examined with a JEM-2100 transmission electron microscope. **c** Schematic diagram of the bacterial binding and synergetic bactericidal mechanism of molecular necklace **1**.

The strong antibacterial activity of the heterometallic necklace **1** was related to severe bacterial cell death. Specifically, confocal observation of the DAPI/propidium iodide (PI) double stained cells revealed that the necklace **1** induced severe cell aggregation and caused almost thorough cell death (indicated by the PI-positive cells with red fluorescence) (Fig. 5f). In contrast, [3] catenane **2** and the metallacycles **3–6** did not cause obvious cell aggregation and only led to partial cell death (Fig. 5f). Similarly, SEM observation showed that heterometallic necklace **1** resulted in much more severe corruption of the cell structures than other complexes (Fig. 5g), confirming drastic cell death induced by necklace **1**. Interestingly, the cells of each group shared the similar levels of intracellular reactive oxygen species (ROS) (Supplementary Fig. 33), excluding the possible contribution of ROS accumulation to the difference in the antibacterial ability of all tested species.

The contribution of cell wall/plasma membrane disruption in bactericidal activity of the necklace **1** was further confirmed by cell surface distribution of the necklace **1** and ultrathin section observations of the bacterial cells (Fig. 6a, b). Energy dispersive spectroscopy mapping showed that both Cu and Pt were abundantly accumulated on the cell surface and co-localized with the cellular elements (e.g., O, N, and P) (Fig. 6a), indicating

the direct contact between the necklace and the cell surface. TEM observations of the bacterial ultrathin sections further revealed that the control cells had intact cell wall and plasma membrane embracing the cytoplasm (Fig. 6b). In contrast, most of the necklace-treated cells experienced severely cell wall/plasma membrane disruption and consequent cytoplasm leakage from the disrupted sites (Fig. 6b), emphasizing the critical role of direct contact between the necklace **1** and the bacterial cell wall/plasma membrane components in bacterial death. In addition, severe DNA fragmentation was observed in necklace **1**-treated cells (Supplementary Fig. 27), implying that DNA cleavage caused by internalized necklace **1** was also involved in its antibacterial performance. Taken together, these results revealed that the heterometallic necklace **1** possessing more positive charges and two metal centers featured much stronger bacterium-binding activity and superior cell wall/plasma membrane/DNA-disrupting capacity than the control complexes, which might be mainly attributed to the synergistic combination of heterometals in the necklace (Fig. 6c).

The stability of necklace **1** and other assemblies in bacterial culture media is certainly worth considering. We then studied their stabilities under a variety of conditions by in situ $^1$H NMR and $^{31}$P NMR spectroscopy. The preliminary results

demonstrated that all assemblies including necklace **1** are relatively stable towards different pH and tryptone, which is used as a nitrogen source in culture media, as indicated by the unchanged $^1$H NMR and $^{31}$P NMR spectra after adding N,N-Diisopropylethylamine, triflic acid (TfOH), or tryptone (Supplementary Figs. 37–40). However, all assemblies would be disassembled in Luria–Bertani (LB) medium because their $^{31}$P NMR spectra became complicated when adding LB medium after 1 h (Supplementary Fig. 41). For comparison, the time dependent $^{31}$P NMR spectra of necklace **1** and metallacycle **4** were recorded and their degradation curves are shown in Supplementary Fig. 42. The results revealed that necklace **1** had a higher resistance to LB medium compared to metallacycle **4**, e.g., on the one hand, the intact metallacycle **4** remained less than 10% of its initial content after 3 min while necklace **1** still kept about 50%; on the other hand, metallacycle **4** was thoroughly destructed within 9 min while a much longer time (~45 min) was needed for necklace **1** to be completely degraded. The enhancement of the stability towards LB medium for necklace **1** was probably due to its aggregation, which decreases its exposed molecular surface area and protects it from LB medium. The stability of coordination-driven self-assembly is essentially related to the intrinsic dynamic nature of metal-ligand coordination chemistry[75]. Based on large amounts of investigation and research on stimuli-responsive metal–ligand assemblies[76,77], we believe that the degradation of necklace **1** probably stems from the ions such as chloride ion in LB medium (Supplementary Figs. 43–50). Obviously, coordination-driven formation of necklace **1** architecture rendered the molecule more stable in biological systems than other metallacycles, and hence prolonged the time of necklace **1**-bacterium interaction to sufficiently kill bacterial cells.

Together, these results indicate that Cu(I) and Pt(II) in necklace **1** in combination play essential roles in bacterial killing: (1) Cu(I) functions in bacterial binding to the cell wall/plasma membrane and disrupts their ultra-structures, together with in promoting DNA cleavage; (2) Pt(II) enhances the biomolecule-interaction capacity of Cu(I) to more efficiently disrupt targeted cell components, e.g., lipopolysaccharides, phospholipids, and DNA; (3) Stoichiometric Cu(I)-contained donor and the Pt(II)-contained acceptor can be precisely integrated into the architecture of necklace **1**, which thus enhances the bacterial binding/damaging capacity and stability of the necklace.

## Discussion

In summary, a heterometallic triangular necklace **1** was successfully synthesized through a "threading-followed-by-ring-closing" approach driven by coordination interaction. The crystal structure of **1** disclosed an elegant triangular necklace architecture containing a large [3 + 3] Pt(II)-N cationic metallacycle interlocked with three polyether phenanthroline macrocycles. The existence of two different metal centers not only facilitated the successful construction of necklace **1** but also endowed it with superior nuclease properties and antibacterial activities. Our studies revealed that the self-assembly of heterometallic necklace would significantly enhance its bactericidal activity. This enhancement might be mainly attributed to the synergistic effect of heterometals in the necklace, endowing it superior bacterium-binding and cell wall/plasma membrane-disrupting capacity for killing the bacterial cells. "Threading-followed-by-ring-closing" approach combining with coordination-driven self-assembly would allow us to further construct more complicated molecular necklaces in the future. And the promising DNA cleavage and antibacterial activities results obtained herein would also attract broad interests and provide directions for future chemical design in this field.

## Methods

The synthesis and characterization of new compounds present in this work, and the experimental details and additional data of DNA cleavage, bacterium-binding and antimicrobial tests are described in the Supplementary Information.

## Data availability

The data that support the findings of this study are available from the authors on reasonable request, see author contributions for specific data sets. The X-ray crystallographic coordinates for structures reported in this study have been deposited at the Cambridge Crystallographic Data Center (CCDC), under deposition numbers CCDC 1889123 (**D1**) and 1889124 (**1**). These data can be obtained free of charge from The Cambridge Crystallographic Data Center via www.ccdc.cam.ac.uk/data_request/cif.

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

## Acknowledgements

H.-B.Y. thanks NSFC/China (Nos. 21572066 and 21625202), Innovation Program of Shanghai Municipal Education Commission (No. 2019-01-07-00-05-E00012), and Program for Changjiang Scholars and Innovative Research Team in University for financial support. X.S. acknowledges the financial supports sponsored by Shanghai Sailing Program (19YF1412900) and the Fundamental Research Funds for the Central Universities. Q.Y. thanks NSFC/China (Nos. 31870139). We thank Dr. Yiwen Wang and Dr. Bing Ni for the TEM study.

## Author contributions

H.-B.Y., X.S., Q.Y., and G.-Y.W. conceived the project, analyzed the data, and wrote the manuscript. G.-Y.W. performed the most of experiments. H.P., H.Q., and X.-L.Z. conducted single crystal analyses. Y.-X.Hu., G.-Q.Y., X.L., and L.X. helped in experiments and data analyses.

## Competing interests

The authors declare no competing interests.
