## [Peer Review File · Nature Communications]

Reviewers' comments:

Reviewer #1 (Remarks to the Author):

This is an important paper presented Dr. Yang and coworkers about the development of novel supramolecular heterometallic triangular necklace and the study of their antibacterial activity. The authors demonstrate that: 1) the five different components can spontaneously assemble into a triangular necklace via coordination-driven self-assembly; 2) These mechanically interlocked necklace are formed in a high yield; 3). These structures show promising antibacterial activity. The authors have done an excellent job on characterizing the supramolecular structures using 1D and 2D NMR, MS, and single crystal X-ray. The studies about the antibacterial activities of such novel compounds are unprecedented. All these findings are novel within the field of metal-based supramolecular chemistry. This work will attract great attentions from the scientific community of supramolecular chemistry and others. Therefore, I recommend the acceptance of this paper. Below is a minor suggestion to improve the manuscript:

1. The compound is stable in organic solvent during NMR studies. But what about its stability in bacteria culture media? Tryptone in the media is very likely to destroy the assembly, since they contain histidine and cysteine. Are these necklaces forming aggregation to enhance their stability and facilitate their bindings to bacteria membrane?

Reviewer #2 (Remarks to the Author):

Review on manuscript NCOMMS-19-32134:

Key aspects: In the manuscript titled "Efficient Self-Assembly of Heterometallic Triangular Necklace with Strong Antibacterial Activity" by Hai-Bo Yang et al. the authors describe the first synthesis of a sophisticated heterometallic mechanically interlocked [4]catenane architecture, which they call molecular necklace. The synthetic approach is a coordination driven self-assembly, by careful choice of orthogonally compatible metals (which do not interfere with each other at the pseudorotaxane/metallacycle formation). The authors were able to isolate and characterize the mechanically-interlocked supramolecule and also were able to characterize it exhaustively including single-crystal X-ray diffraction. The latter is usually very challenging to achieve for such sophisticated architectures and is certainly an outstanding feature of this publication. The [4]catenane then was tested towards applicability in the biomedical context, in detail DNA cleavage activity as well as antibacterial performance was evaluated and compared with analogous compounds. The authors identified a superior antibacterial activity, which they attribute to a synergistic effect of the two metal ions being present in the [4]catenane.

Validity: The manuscript combines both the synthetic aspects of challenging supramolecular/mechanically interlocked architectures with the applicability of the metal-based

supramolecules in the biological context – showing a high antibacterial activity - which certainly qualifies the work to be published in a journal with a general readership as Nature Communications. The authors nicely present the synthesis of the molecular necklace and unambiguously prove its formation. However, the second part of the manuscript, i.e. the application in the biomedical context falls a bit short mainly due to a missing conceptual link between the synthetic part and the application part. In the first part, the authors explain that the main challenge of the published work to synthesize the molecular architecture was to carefully choose a metal ion combination which does not interfere during the metallamacrocycle formation. As a note, this should be more elaborated by the authors, including possible mistakes of metal combination or at least reasoning, why this is actually a challenge. In any case, following this argumentation, the conception of the supramolecule was purely synthetically driven. However, in the second part of the manuscript the 'synergistic' combination of Pt and Cu is used to explain the superior antibacterial activity of the necklace compared to other tested compounds.

Now, even if the choice of combination of the metals in the necklace was not based on a targeted biomedical application during the conception process and the bioactivity was a real discovery later on, I am missing a more detailed rationalization of the observed results. What do the authors think is the role of Pt, what is the role of Cu? In case of the reference the authors cite [49-52] also heterometallic complexes featuring Pt and Cu have been used, in which both metals were attributed a certain function (Cu mimicking hydrogenase activity, Pt mimicking cis-platinum-DNA-interaction). Do the authors think this is the same in their case? Is it only the combination of the metals in any ligand platforms which determines the activity, or why do the authors think their complex architecture might be advantageous? Also, if it is really the architecture (which interacts e.g. via stacking etc.), do the authors think the metals are the decisive components? If so, do the authors think, the metals do dissociate under physiological conditions and therefore present e.g. DNA hydrolysis activity. Do the authors think, the necklace stays intact during the DNA cleavage and did they maybe test the stability at different pH or ion concentrations (e.g. chloride)?

I think, the authors made already a series of very nice experiments and have already highly promising results concerning the biological activity. I also understand, that the format of a communication has been chosen to publish this highly interesting initial results. However, a more detailed conceptual reasoning of the novelty, possible superiority and significance of the supramolecular/MIM approach for this kind of metallodrugs is certainly advised before accepting the manuscript for a journal like Nature Communications.

Data and methodology: I specifically focus on the synthesis and characterization part of the manuscript, since the biological testing is outside of my expertise. The authors are certainly experts on the area of coordination driven supramolecular self-assembly and the results are presented in a correct and graphically understandable manner.

The refined models from the SC-XRD measurements have issues, which should be resolved before publication. For the refinement of crystal structure of D1 (exp_571) a huge number (> 100!!) of reflections have been omitted during refinement. Why is this? Usually reflections should be only

omitted, if they are systematically affected (e.g. covered by the beam stop) and not to 'improve' the data set quality. Furthermore, the PLATON-SQUEEZE/solvent mask procedure has been applied (ABIN card can be found in the refinement instructions) This should be only done, if defined voids are present in the crystal structure, which contain (disordered) solvent molecules which cannot be refined properly. The reason and amount of removed solvent molecules should additionally be properly reported. This should be done in the present case.

In the second model, there are apparent issues. A and B alerts in the checkcif routine have to be resolved. There seems also be an issue with the overall molecular sum formula. Also, when displaying the refined model, there are Pt atoms at different positions, but not refined correctly. The model has to be significantly improved, before it can be accepted for publication.

As a remark on the discussion of the single-crystal X-ray structure: The authors mention the existence of two enantiomers in both of the crystal structures. Naively speaking, this is the result of the centrosymmetric space group, in which the respective compound is crystallizing. This is practically the case for all molecules, e.g. also conformational isomers. So in the solid state, they would also behave like mirror images, but would not be classified as enantiomers (since in solution they would easily interconvert via rotation of bonds). Likewise, in solution, I would not call compound D1 a chiral compound (in solution), since the two 'mirror images' would also easily interconvert into each other. If the authors really want to emphasize an actual chirality in the solid state, they should prove/elaborate on this. In the current manuscript it feels a bit like a random information, not adding any value to the content.

Conclusions: As stated above, the conclusions drawn by the authors should be put into a broader conceptual context and discussed respectively.

Reviewer #3 (Remarks to the Author):

In this manuscript, Shi, Yu, Yang and coworkers describe the highly efficient self-assembly of heterometallic triangular molecular necklace and its applications in DNA-cleaving and antibacterial. A [3]catenane and a triangular molecular necklace were firstly prepared by using orthogonal combination of Cu(I)-based coordination template method and Pt(II)-based coordination self-assembly. Their structures were well characterized. Especially, single-crystal X-ray analysis unambiguously confirmed the formation of catenated triangular molecular necklace. More interestingly, this molecular necklace showed higher DNA-cleaving and antibacterial activities than its analogues including [3]catenane 2 and

metallacycles 3, 4, 5, and 6. The manuscript is clearly and concisely written and the presented results are great interesting to the broad readership of Nature Communications. Therefore, I highly recommend the publication of this manuscript after the following minor points are addressed:

(1) As Nature Communications has no limitation on pages, it is recommended to add the structures of [3]catenane 2 and metallacycles 3, 4, 5, and 6 into the main text, which will improve the presentation and facilitate reading of the paper.

(2) In the manuscript, the high antibacterial activity of necklace 1 is attributed to its more positive charges and the synergistic effect of heterometals in the necklace. However, as seen in Supplementary Table 1, D1 also has comparatively high antibacterial activity, but a little lower than that of necklace 1, towards the tested pathogens. Is it possible that Cu^+ play significant role on antibacterial? It is recommended to give some clues to explain it.

Response to Reviewer 1:

This is an important paper presented Dr. Yang and coworkers about the development of novel supramolecular heterometallic triangular necklace and the study of their antibacterial activity. The authors demonstrate that: 1) the five different components can spontaneously assemble into a triangular necklace via coordination-driven self-assembly; 2) These mechanically interlocked necklace are formed in a high yield; 3). These structures show promising antibacterial activity. The authors have done an excellent job on characterizing the supramolecular structures using 1D and 2D NMR, MS, and single crystal X-ray. The studies about the antibacterial activities of such novel compounds are unprecedented. All these findings are novel within the field of metal-based supramolecular chemistry. This work will attract great attentions from the scientific community of supramolecular chemistry and others. Therefore, I recommend the acceptance of this paper. Below is a minor suggestion to improve the manuscript:

Reply: We greatly appreciate the reviewer's positive comments on the work in this manuscript.

1. The compound is stable in organic solvent during NMR studies. But what about its stability in bacteria culture media? Tryptone in the media is very likely to destroy the assembly, since they contain histidine and cysteine. Are these necklaces forming aggregation to enhance their stability and facilitate their bindings to bacteria membrane?

Reply: The stability of necklace **1** and other assemblies especially in bacterial culture media is certainly worth considering. According to the reviewer's suggestion, we have studied their stabilities under a variety of conditions by *in situ* ^1H NMR and ^{31}P NMR spectroscopy.

Firstly, the stability of necklace **1** and other assemblies to acid and base was evaluated. As shown in Supplementary Figures 24-29, *in situ* ^1H NMR and ^{31}P NMR spectra of all assemblies remained unchanged after adding six equivalents of N,N-Diisopropylethylamine (DIEA) or triflic acid (TfOH), indicating the good stability of these assemblies at different pH, which is also consistent with the previous

report (*RSC Adv.* 2014, **4**, 2605–2608).

Next, in consideration of the nitrogen source in culture media such as tryptone that may destroy the structure of assemblies, we studied their stability towards tryptone by *in situ* ^{31}P NMR spectrum (Supplementary Figure 30). Similarly, all the ^{31}P NMR spectra of assemblies almost remained the same when tryptone was added. In contrast, in the case of Pt(II)-contained acceptors **A1** and **A2**, new resonance peaks appeared. Thus we infer that above coordination-driven self-assemblies including necklace **1** are relatively stable towards tryptone, while the Pt(II) acceptor is not stable probably due to the amino acid in tryptone.

Then, the stability of all assemblies in LB (Luria-Bertani) medium was tested. The final ^{31}P NMR spectra of all assemblies and acceptors **A1** and **A2** became complicated when adding LB medium after one hour (Supplementary Figure 31), revealing the destruction of molecular frameworks. The time dependent ^{31}P NMR spectra of necklace **1** and metallacycle **4** and their stability curves are shown in Figure R1. The preliminary results revealed that necklace **1** had a higher resistance to LB medium compared to metallacycle **4**, *e.g.*, on the one hand, the intact metallacycle **4** remained less than 10% of its initial content after 3 min while necklace **1** still kept about 50%; on the other hand, metallacycle **4** was thoroughly destructed within 9 min while a much longer time (~ 45 min) was needed for necklace **1** to be completely degraded. The enhancement of the stability towards LB medium for necklace **1** was probably due to its aggregation, which decreases its exposed molecular surface area and protects it from LB medium. Based on the large amounts of investigation and research on stimuli-responsive metal–ligand assemblies (*e.g.* *Chem. Rev.* 2015, **115**, 7729–7793), we believe that the degradation of necklace **1** probably stems from the ions such as chloride ion in LB medium (Supplementary Figures 33–40). The stability of coordination-driven self-assembly is essentially related to the intrinsic dynamic nature of metal-ligand coordination chemistry.

Figure R1. Time dependent ³¹P NMR spectra of metallacycle 4 (a) and necklace 1 (b), and their degradation curves (c). The degradation curve was obtained based on the integration values of ³¹P NMR spectra.

We have also added the stability description and the related data in both revised manuscript and Supporting Information:

“The stability of necklace 1 and other assemblies in bacterial culture media is certainly worth considering. We then studied their stabilities under a variety of conditions by in situ ¹H NMR and ³¹P NMR spectroscopy. The preliminary results demonstrated that all assemblies including necklace 1 are relatively stable towards different pH and tryptone, which is used as a nitrogen source in culture media, as indicated by the unchanged ¹H NMR and ³¹P NMR spectra after adding N,N-Diisopropylethylamine (DIEA), triflic acid (TfOH) or tryptone (Supplementary Figures 24-30)....”

Response to Reviewer 2:

Key aspects: In the manuscript titled “Efficient Self-Assembly of Heterometallic Triangular Necklace with Strong Antibacterial Activity” by Hai-Bo Yang et al. the authors describe the first synthesis of a sophisticated heterometallic mechanically interlocked [4]catenane architecture, which they call molecular necklace. The synthetic approach is a coordination driven self-assembly, by careful choice of orthogonally compatible metals (which do not interfere with each other at the pseudorotaxane/metallacycle formation). The authors were able to isolate and characterize the mechanically-interlocked supramolecule and also were able to characterize it exhaustively including single-crystal X-ray diffraction. The latter is usually very challenging to achieve for such sophisticated architectures and is certainly an outstanding feature of this publication. The [4]catenane then was tested

towards applicability in the biomedical context, in detail DNA cleavage activity as well as antibacterial performance was evaluated and compared with analogous compounds. The authors identified a superior antibacterial activity, which they attribute to a synergistic effect of the two metal ions being present in the [4]catenane.

Reply: We greatly appreciate the reviewer's positive comments on the work in this manuscript.

Validity: The manuscript combines both the synthetic aspects of challenging supramolecular/mechanically interlocked architectures with the applicability of the metal-based supramolecules in the biological context – showing a high antibacterial activity - which certainly qualifies the work to be published in a journal with a general readership as Nature Communications. The authors nicely present the synthesis of the molecular necklace and unambiguously prove its formation. However, the second part of the manuscript, i.e. the application in the biomedical context falls a bit short mainly due to a missing conceptual link between the synthetic part and the application part. In the first part, the authors explain that the main challenge of the published work to synthesize the molecular architecture was to carefully choose a metal ion combination which does not interfere during the metallamacrocyclic formation. As a note, this should be more elaborated by the authors, including possible mistakes of metal combination or at least reasoning, why this is actually a challenge. In any case, following this argumentation, the conception of the supramolecule was purely synthetically driven. However, in the second part of the manuscript the 'synergistic' combination of Pt and Cu is used to explain the superior antibacterial activity of the necklace compared to other tested compounds.

Reply: We fully understand the reviewer's concern about the conceptual link between the synthetic part and the application part in this work. In order to address this concern, the logic and conception between coordination complexes and biomedical applications has been well organized in the Introduction section in the revised version, which will hopefully facilitate the readers understand the logic in this article.

It should be noted that the biomedical applications of coordination-driven self-assembly of metallacycles and metallacages have already been systematically investigated. It has been demonstrated that these metallosupramolecular complexes can serve well as anticancer agents, drug delivery systems, biosensors, DNA

intercalators, antibacterial agents and so on (see reviews and articles: *Acc. Chem. Res.* 2013, **46**, 2464–2474; *J. Am. Chem. Soc.* 2019, **141**, 14005–14020; *Nat. Commun.* 2018, **9**, 1815; *Proc. Natl. Acad. Sci. U. S. A.* 2019, **116**, 23437–23443). Inspired by these reports, the potential biomedical applications of the novel coordination-driven self-assembly of necklaces presented in this study are highly expected. Moreover, given the challenging synthesis and rare applications of mechanically-interlocked molecules, the obtained necklace in this study may represent the first example of mechanically-interlocked molecule for biomedical applications, which will attract broad interests in the field of supramolecular chemistry and biomaterial science.

On the other hand, the emerging drug-resistant bacterial pathogens provoke the development of novel antibacterial agents. Owing to the significance of bacterial DNA and cell wall/plasma membrane in survival and growth of the bacterial cells, one important strategy against the bacteria will be how to enhance the DNA/cell wall/plasma membrane-disrupting capacity of the antibacterial agents with high stability. The coordination-driven self-assembly of the positively charged metallosupramolecular complexes just provides a powerful platform to enhance the electric charges for severely disrupting of the targeted DNA/cell wall/plasma membrane, and to improve the stability of the necklaces for prolonging the interaction time between the necklaces and the bacterial cells. Again, inspired by the recent development of biomedical and biochemical applications of coordination-driven self-assembly, we envision that development of molecular necklaces possessing multiple metals as well as the enhanced electric charges and improved stability may potentially serve as efficient DNA intercalators and bacterial cell wall/plasma membrane-disrupting agents, thus opening a new window to their biomedical applications.

Following the reviewer's advice, we have added some conceptual introduction to explain the objective and significance of this work:

In Introduction section: ***“It's worth noting that, though a major breakthrough in molecular necklaces regarding to their impressive structures as well as well-developed synthetic strategy has been achieved, the study of their properties and potential applications have persistently lagged behind expectations. To the best of our knowledge, the focus of the chemistry of MIMs, especially for molecular***

necklaces, has still stayed on molecular design and synthesis stage....

The emerging drug-resistant bacterial pathogens have been becoming great threats to our human health, and thus provoke the development of novel antibacterial agents^{47,48}. The bacterial DNA, cell wall, and plasma membrane are critical components of bacterial cells essential for survival and growth⁴⁹. The negatively charged property of DNA, bacterial cell wall (displaying negatively charged lipopolysaccharides or teichoic acid), and plasma membrane (exposing negatively charged phospholipids)⁵⁰ indicates that they are realistic targets of the positively charged metal coordinating complexes....”

In DNA-cleaving and antibacterial activities section: *“DNA, a double helix carrying the genetic instructions, is of high importance for most of organisms. DNA is the primary target for the most anticancer drugs and anti-infection therapies⁵⁸. Investigations on DNA cleavage and developments of novel chemical nucleases have attracted wide interest owing to their potential applications as new therapeutic agents as well as new diagnostic structural probes to analyze DNA information....*

Strong DNA cleavage activity of the coordination complexes commonly indicates high antibacterial ability⁶⁸⁻⁷⁰. Moreover, recent studies have further unveiled some interesting findings that metallacycles and metallacages exhibited strong cell wall (together with plasma membrane)-intercalating ability and antibacterial activity^{44-46,71}. Considering the fact that molecular necklace 1 has combinational positive charges, π - π stacking ability, platinum(II)-pyridine coordination and $[\text{Cu}(\text{phen})_2]^+$ units, we speculate that it may possess strong antibacterial activity.”

In order to elaborate the challenge of the synthesis of heterometallic triangular necklace, some additional controlled experiments were carried out. For instance, another molecular necklace **7** through coordination-driven self-assembly between 3,8-disubstituted $[\text{Cu}(\text{phen})_2]^+$ and the corresponding diplatinum (II) acceptors (Figure R2) was proposed. However, the complicated ^1H NMR and ^{31}P NMR spectrum indicated the unsuccessful self-assembly process (Supplementary Scheme 9 and Supplementary Figure 10). This indicates that not all the ligands are applicable to self-assemble to form the designed necklace, thus proving the challenge to synthesize such heterometallic molecular necklace.

We realized that two metal centers would result in two distinct different

coordination motifs, and thus compatibility issues may arise. Actually, we found that the compatibility of the selected two coordination motifs was of vital importance for the successful construction of molecular necklaces. For example, 2,9-disubstituted $[\text{Cu}(\text{phen})_2]^+$ and platinum(II)–pyridine coordination motifs were demonstrated to be mutually compatible and would not interfere each other during the formation of metallacycle, while these two coordination motifs, 3,8-disubstituted $[\text{Cu}(\text{phen})_2]^+$ and platinum(II)–pyridine, were not compatible because of their potential competing coordination reaction. We suspect that the steric hindrance of 2,9-disubstituted $[\text{Cu}(\text{phen})_2]^+$ may well protect $[\text{Cu}(\text{phen})_2]^+$ from diplatinum (II) acceptor (Figure R2).

Figure R2. Unsuccessful attempt for the synthesis of molecular necklace 7 and metallacycle 8.

We have then added the corresponding description and the unsuccessful trial in both revised manuscript and Supporting Information:

“In order to elaborate the challenge of the synthesis of heterometallic triangular necklace, some additional controlled experiments were carried out. For instance, another molecular necklace 7 through coordination-driven self-assembly between 3,8-disubstituted $[\text{Cu}(\text{phen})_2]^+$ D5 and the corresponding diplatinum (II) acceptors A3 was proposed (Supplementary Scheme 9). However, the complicated ^1H NMR and ^{31}P NMR spectrum indicated the unsuccessful self-assembly process....”

Now, even if the choice of combination of the metals in the necklace was not based on a targeted biomedical application during the conception process and the bioactivity was a real discovery later on, I am missing a more detailed rationalization of the observed results. What do the authors think is the role of Pt, what is the role of Cu? In

case of the reference the authors cite [49-52] also heterometallic complexes featuring Pt and Cu have been used, in which both metals were attributed a certain function (Cu mimicking hydrogenase activity, Pt mimicking cis-platinum-DNA-interaction). Do the authors think this is the same in their case? Is it only the combination of the metals in any ligand platforms which determines the activity, or why do the authors think their complex architecture might be advantageous? Also, if it is really the architecture (which interacts e.g. via stacking etc.), do the authors think the metals are the decisive components? If so, do the authors think, the metals do dissociate under physiological conditions and therefore present e.g. DNA hydrolysis activity. Do the authors think, the necklace stays intact during the DNA cleavage and did they maybe test the stability at different pH or ion concentrations (e.g. chloride)?

Reply: We fully understand the reviewer's concern about the role of both Pt and Cu in biological application in this work. Our preliminary results have revealed that the presence of the two metal ions and the architecture of positively charged molecular necklace have significant impact on the antibacterial performance of necklace **1**. On the one hand, the metals, especially Cu, play essential roles in disrupt cellular components, and the synergy of Cu and Pt might contribute to the highest capacity of necklace **1** to disrupt the cell components (e.g., cell wall/plasma membrane, DNA). On the other hand, the positively charged architecture of molecular necklace **1**, which has a coordination-based molecular framework and the two metal ions, might possess antibacterial activity owing to its strong binding with the negatively charged components of the cell wall/plasma membrane, e.g., liposaccharides (LPS) and phospholipids.

To verify these hypotheses, a series of experiments were performed to elucidate the contribution of the metal ions and the architecture of necklace **1** to its antibacterial performance. There is no doubt that both metal ions have important roles in the antibacterial process. For Cu(I), all of the Cu(I)-containing molecules (i.e., **1**, **2**, **D1**, and Cu(MeCN)₄PF₆) had higher antibacterial activity (Figure 4d, Supplementary Figure 20, Table S1), and have remarkably stronger LPS-binding and membrane-damaging ability than the Cu(I)-free molecules (i.e., **3**, **4**, **5**, **6**, **A1**, and **A2**) (Supplementary Figure 21). These results indicated that Cu(I) plays an important role in killing bacterial cells by enhancing the interaction between **1** and the cell membrane components. Moreover, the Cu(I)-containing molecules (i.e., **1**, **2**, **D1**) had

DNA cleavage activity, while the Pt(II)-containing but Cu(I)-free molecules (*i.e.*, **3**, **4**, **5**, **6**, **A1**, and **A2**) did not show this activity, indicating that Cu(I) rather than Pt(II) functions as DNA nuclease to induce DNA cleavage. For Pt(II), all of the DNA cleavage and antibacterial experiments showed that only the co-existence of Pt(II) and Cu(I) led to the highest DNA cleavage and antibacterial activity, implying that Pt(II) might enhance the biomolecule-disruption capacity of Cu(I). Therefore, Cu(I) and Pt(II) had a combinational effect on killing bacterial cells.

The excellent bacterial binding ability of necklace **1** architecture has been successfully confirmed by the glass-adhering test, dynamic light scattering (DLS) analysis, and liposome damage assay (Figure 4a, Supplementary Figure 19). In order to further provide insight into the importance of necklace architecture in bacterial killing process, some additional control experiments have been further carried out. For example, the antibacterial activities of the ligand mixtures of **D1+A1'** or **D1+A2'** have been surveyed (Figure R3). Notably, the physical blends of **D1+A1'** or **D1+A2'** could not self-assemble into the necklace since Pt was protected by iodide or bromide moieties. It was found that, compared to the necklace **1**, the mixture of **D1+A1'** or **D1+A2'** exhibited much weaker bacterium-killing efficiency (Figure R3). These results implied the significance of necklace architecture in bacterial killing.

We have added the corresponding description and related figures in both revised manuscript and Supporting Information:

*“Meanwhile, the Cu(I)-containing molecule D1, rather than the Cu(I)-free molecules (*i.e.*, A1, A2, 3, 4, 5 and 6), also led to obvious DNA degradation, but its DNA cleavage efficiency was much lower than the necklaces (Supplementary Figure 15). This observation indicated that Cu(I) plays a critical role in DNA cleavage by the necklaces, and its efficiency was remarkably enhanced by the Pt(II)-containing acceptor.”*

*“An interesting observation is that all of the Cu(I)-containing molecules (*i.e.*, 1, 2, D1, and $\text{Cu}(\text{MeCN})_4\text{PF}_6$) had higher antibacterial activity than the Cu(I)-free molecules (*i.e.*, 3, 4, 5, 6, A1, and A2) (Figure 4d, Supplementary Figure 20, Table S1). In addition, we also found that all of the Cu(I)-containing molecules have remarkably stronger LPS-binding and membrane-damaging ability than the other molecules (Supplementary Figure 21). These results indicated that Cu(I) plays an*

important role in killing bacterial cells by enhancing the interaction between 1 and the cell membrane components. Notably, only the cooperation of the Cu(I)-contained donor and the Pt(II)-contained acceptor in the necklace 1 generated the highest cell wall/plasma membrane-binding/damaging and bacterial killing ability since the Cu(I)-free matallacycle 4 and the Pt(II)-free donor D1 exhibited much lower antibacterial capacity than necklace 1 (Figure 4, Supplementary Figure 20).”

“Coordination-driven formation of the necklace 1 framework is also critical for efficiently killing the pathogens. To prove this contribution, a control experiment was carried out. The antibacterial activities of the ligand mixtures of D1+A1’ or D1+A2’ have been surveyed. Notably, the physical blends of D1+A1’ or D1+A2’ could not self-assemble into the necklace since Pt was protected by iodide or bromide moieties. It was found that, compared to the necklace 1, the mixtures of D1+A1’ or D1+A2’ exhibited much weaker bacterium-killing efficiency (Supplementary Figure 22). These results further implied the significance of necklace formation in bacterial killing. From the above discussion, we may conclude that stoichiometric Cu(I)-contained donor and the Pt(II)-contained acceptor can be precisely integrated into the platform of necklace 1, which is highly conducive to its outstanding bacterial killing ability. ”

Figure R3. Cell viability of the groups treated by the necklace 1 (8 μ M), the donor D1 (8 μ M), the analogue of A1 (A1’, 8 μ M), the A2 analogue of (A2’, 8 μ M), the mixture of D1 and A1’ (D1+A1’, 8 μ M + 8 μ M), and the mixture of D1 and A2’ (D1+A2’, 8 μ M + 8 μ M). Different letters above the columns indicate significant difference between the groups ($P < 0.05$).

In addition, more additional experiments of energy dispersive spectroscopy (EDS) mapping and transmission electron microscopy (TEM) were performed to examine the morphological changes of necklace-treated bacterial cells. EDS mapping showed that both Cu and Pt were abundantly accumulated on the cell surface (Figure R4a), indicating the direct contact between the necklace and the cell surface. The electron microscopy of the ultrathin sections of necklace-treated bacterial cells revealed that necklace **1** possessing positive charges and two metal centers induced severely cell wall/plasma membrane disruption and consequent cytoplasm leakage from the disrupted sites (Figure R4b). All results revealed that the heterometallic necklace **1** possessing more positive charges and two metal centers featured much stronger bacterium-binding activity and superior cell wall/plasma membrane-disrupting capacity than the controlled complexes, which might be mainly attributed to the synergistic effect of heterometals in the necklace. We believed that the architectures of such coordination-driven self-assemblies like necklace **1** have several advantages including the highly dynamic metal–ligand coordination bonds, diverse metal ions, positively charged molecular nature, together with their highly tunable coordination geometries, which would endow them with great merits in biomedical applications. In addition, necklace **1** could serve as a good platform where stoichiometric Cu and Pt ions can be precisely integrated, which is highly conducive to its outstanding bacterial killing ability. Based on all above results, we inferred that the architecture of **1** would strongly bind the cell wall/plasma membrane, and then disrupt the membrane probably because of the presence of the metal ions.

Figure R4. Contribution of bacterial binding and cell wall/plasma membrane disruption to bactericidal activity of the necklace **1**. (a) EDS mapping of the main elements (*i.e.*, O, N, P, Cu and Pt) on the bacterial cell surface. (b) TEM observation of ultrathin sections of the control and necklace **1**-treated cells (after 30 min of treatment). The red, yellow, and blue arrows indicate cell wall (CW) disruption, plasma membrane (PM) disruption, and cytoplasm leakage, respectively. Ultrathin sections were cut with a diamond knife, collected on 200-mesh copper grids, stained with uranyl acetate and lead citrate, and examined with a JEM-2100 transmission electron microscope

We have added the corresponding description and related figures in the revised manuscript:

“The contribution of cell wall/plasma membrane disruption in bactericidal activity of the necklace 1 was further confirmed by cell surface distribution of the necklace 1 and ultrathin section observations of the bacterial cells (Figure 5a, 5b). Energy dispersive spectroscopy (EDS) mapping showed that both Cu and Pt were abundantly accumulated on the cell surface and co-localized with the cellular elements (e.g., O, N, and P) (Figure 5a), indicating the direct contact between the necklace and the cell surface....”

Regarding the stability concern of the assemblies including necklace **1**, their stabilities under a variety of conditions including acid, base, tryptone (which is used as a nitrogen source in culture media), LB (Luria-Bertani) medium and chloride ion have been thoroughly surveyed (see Supplementary Figures 24-40 in the revised Supporting Information). The obtained results demonstrated that all assemblies including necklace **1** are relatively stable towards different pH and tryptone, which is used as a nitrogen source in culture media, but would be disassembled in LB medium or in the presence of chloride ion because of the dynamic nature of metal–ligand bonds. The released Pt from necklace **1** might further enter into the bacterial cells and caused DNA cleavage, which was confirmed by DNA fragmentation in the necklace **1**-treated cells. Moreover, the comparison experiment showed that necklace **1** had a higher resistance to LB medium probably due to its aggregation that decreased its exposed molecular surface area and protected it from LB medium.

We have added the stability description and related data, and summarized the roles of Cu(I) and Pt(II) in both revised manuscript and Supporting Information:

“The stability of necklace 1 and other assemblies in bacterial culture media is certainly worth considering. We then studied their stabilities under a variety of conditions by in situ ¹H NMR and ³¹P NMR spectroscopy. The preliminary results demonstrated that all assemblies including necklace 1 are relatively stable towards different pH and tryptone, which is used as a nitrogen source in culture media, as indicated by the unchanged ¹H NMR and ³¹P NMR spectra after adding N,N-Diisopropylethylamine (DIEA), triflic acid (TfOH) or tryptone (Supplementary Figure 24-30)....”

“Together, these results indicate that Cu(I) and Pt(II) in necklace 1 in combination play essential roles in bacterial killing: (1) Cu(I) functions in bacterial binding to the cell wall/plasma membrane and disrupt their ultra-structures, together with in promoting DNA cleavage; (2) Pt(II) enhances the biomolecule-interaction capacity of Cu(I) to more efficiently disrupt targeted cell components, e.g., lipopolysaccharides, phospholipids, and DNA; (3) Stoichiometric Cu(I)-contained donor and the Pt(II)-contained acceptor can be precisely integrated into the architecture of necklace 1, which thus enhances the bacterial binding/damaging

capacity and stability of the necklace.”

I think, the authors made already a series of very nice experiments and have already highly promising results concerning the biological activity. I also understand, that the format of a communication has been chosen to publish this highly interesting initial results. However, a more detailed conceptual reasoning of the novelty, possible superiority and significance of the supramolecular/MIM approach for this kind of metallodrugs is certainly advised before accepting the manuscript for a journal like Nature Communications.

Reply: We completely agree with the reviewer’s opinion on the current manuscript. According to the reviewer’s suggestion, the logic and conception between coordination complexes and biomedical applications has been well organized in the revised version. Moreover, some additional control experiments have been carried out, which provide more supports to such conceptual link between the synthetic part and the biological application. We believe that, in this work, besides the challenging synthesis and systematically structural characterizations, the DNA cleavage and antibacterial results of supramolecular necklace may represent the first example of mechanically-interlocked molecule that possesses biomedical applications, which will attract broad interests in the field of supramolecular chemistry and biomaterial science. We hope that this study could make some contribution to understand the possibility/potential of the application of the supramolecular/MIM approach in biological application.

Data and methodology: I specifically focus on the synthesis and characterization part of the manuscript, since the biological testing is outside of my expertise. The authors are certainly experts on the area of coordination driven supramolecular self-assembly and the results are presented in a correct and graphically understandable manner.

The refined models from the SC-XRD measurements have issues, which should be resolved before publication. For the refinement of crystal structure of D1 (exp_571) a huge number (> 100!!) of reflections have been omitted during refinement. Why is this? Usually reflections should be only omitted, if they are systematically affected (e.g. covered by the beam stop) and not to ‘improve’ the data set quality. Furthermore, the PLATON-SQUEEZE/solvent mask procedure has been applied (ABIN card can be found in the refinement instructions) This should be only done, if defined voids are

present in the crystal structure, which contain (disordered) solvent molecules which cannot be refined properly. The reason and amount of removed solvent molecules should additionally be properly reported. This should be done in the present case.

In the second model, there are apparent issues. A and B alerts in the checkcif routine have to be resolved. There seems also be an issue with the overall molecular sum formula. Also, when displaying the refined model, there are Pt atoms at different positions, but not refined correctly. The model has to be significantly improved, before it can be accepted for publication.

Reply: In the first model, according to the reviewer's suggestion, the crystal structure of D1 (exp_571) was refined.

In the second model, the asymmetric unit is the whole triangular necklace, leading to large unit cell and significant numbers of atoms in it, not to mention the disordered solvents in the void being removed by SQUEEZ routine. As the results, the reflection numbers total is also very large. Therefore, having a large number of reflections omitted is not abnormal.

We have included this in the main text: ***“The compound crystallizes in P-1 space group with the asymmetric unit is the whole triangular necklace 1 molecule. The large void inside the triangular necklace is filled with disordered solvents, which are removed using SQUEEZ routine of the Platon program during the crystal structure refinement.”***

All B alerts are due to some groups/atoms in the outer layers of the triangular necklace being very disordered. This could be the results of those atoms/groups being disordered themselves or caused by disordered solvents in the void. Some H-atoms attaching to disordered C-atoms shift during refinement. Usually, we can add some constraints to limit their movements and have a satisfactory cif file. In this case, however, the refinement process is VERY slow due to very large structure, so we chose to omit these H-atoms. This does not affect to the total quality of the structure. The bond precision for the core of the triangular necklace is good enough for analysis.

The revised cif and checkcif files of both models have been updated in the resubmitted file.

As a remark on the discussion of the single-crystal X-ray structure: The authors

mention the existence of two enantiomers in both of the crystal structures. Naively speaking, this is the result of the centrosymmetric space group, in which the respective compound is crystallizing. This is practically the case for all molecules, e.g. also conformational isomers. So in the solid state, they would also behave like mirror images, but would not be classified as enantiomers (since in solution they would easily interconvert via rotation of bonds). Likewise, in solution, I would not call compound D1 a chiral compound (in solution), since the two ‘mirror images’ would also easily interconvert into each other. If the authors really want to emphasize an actual chirality in the solid state, they should prove/elaborate on this. In the current manuscript it feels a bit like a random information, not adding any value to the content.

Reply: We have carefully looked through the literature about the chirality of $[\text{Cu}(\text{phen})_2]^+$ and confirmed that the pseudo-tetrahedral $[\text{Cu}(\text{phen})_2]^+$ has intrinsic chirality but undergoes rapid ligand exchange in solution (*Dalton Trans.* 2017, **46**, 6553–6569; *Dalton Trans.* 2006, 2058–2065; *Inorg. Chem.* 1998, **37**, 2145-2149; *Chem. - A Eur. J.* 2015, **21**, 8851–8858; *Chem. Soc. Rev.* 2018, **47**, 5266–5311). We fully agree with the reviewer that the chirality of molecular necklace presented in this work is a casual discovery that may not closely relate to its biomedical properties and applications. At this stage it may be implausible to separate and systematically investigate the two enantiomers herein; however, we hope that the casual discovery of the chirality in molecular necklace would enrich the recent development of chiral chemistry in MIMs such as rotaxanes and catenanes and might be useful to some readers working in this research area.

Conclusions: As stated above, the conclusions drawn by the authors should be put into a broader conceptual context and discussed respectively.

Reply: According to the reviewer’s suggestion, we have made the corresponding changes in the final Discussion section as follows:

“...This enhancement might be mainly attributed to the synergistic effect of heterometals in the necklace, endowing it superior bacterium-binding and cell wall/plasma membrane-disrupting capacity for killing the bacterial cells. “Threading-followed-by-ring-closing” approach combining with coordination-driven self-assembly would allow us to further construct more

complicated molecular necklaces in the future. And the promising DNA cleavage and antibacterial activities results obtained herein would also attract broad interests and provide directions for future chemical design in this field.”

Response to Reviewer 3:

In this manuscript, Shi, Yu, Yang and coworkers describe the highly efficient self-assembly of heterometallic triangular molecular necklace and its applications in DNA-cleaving and antibacterial. A [3]catenane and a triangular molecular necklace were firstly prepared by using orthogonal combination of Cu(I)-based coordination template method and Pt(II)-based coordination self-assembly. Their structures were well characterized. Especially, single-crystal X-ray analysis unambiguously confirmed the formation of catenated triangular molecular necklace. More interestingly, this molecular necklace showed higher DNA-cleaving and antibacterial activities than its analogues including [3]catenane 2 and metallacycles 3, 4, 5, and 6. The manuscript is clearly and concisely written and the presented results are great interesting to the broad readership of Nature Communications. Therefore, I highly recommend the publication of this manuscript after the following minor points are addressed:

Reply: We greatly appreciate the reviewer’s positive comments on the work in this manuscript.

(1) As Nature Communications has no limitation on pages, it is recommended to add the structures of [3]catenane 2 and metallacycles 3, 4, 5, and 6 into the main text, which will improve the presentation and facilitate reading of the paper.

Reply: Following the reviewer’s advice, we have added the structures of [3]catenane 2 and metallacycles 3, 4, 5, and 6 into the main text in the revised manuscript as shown in Scheme 1.

(2) In the manuscript, the high antibacterial activity of necklace 1 is attributed to its more positive charges and the synergistic effect of heterometals in the necklace. However, as seen in Supplementary Table 1, D1 also has comparatively high antibacterial activity, but a little lower than that of necklace 1, towards the tested pathogens. Is it possible that Cu⁺ play significant role on antibacterial? It is recommended to give some clues to explain it.

Reply: We do agree with the reviewer that Cu^+ plays a significant role on antibacterial application in this study. Similar to the response to the second reviewer, some additional experiments have been carried out to demonstrate the importance of Cu and the possible synergistic effect of heterometals in the necklace. For example, we observed that all of the Cu(I)-containing molecules (*i.e.*, **1**, **2**, **D1**, and $\text{Cu}(\text{MeCN})_4\text{PF}_6$) had a higher antibacterial activity than the Cu(I)-free molecules (*i.e.*, **3**, **4**, **5**, **6**, **A1**, and **A2**) (Figure 4d, Supplementary Figure 20, Table S1). In addition, we also found that all the Cu(I)-containing molecules have remarkably stronger LPS-binding and membrane-damaging ability than the other molecules (Supplementary Figure 21). These results indicated that Cu(I) plays an important role in killing bacterial cells by enhancing the interaction between assemblies and the cell membrane components. More importantly, we have also demonstrated that only the cooperation of the Cu(I)-contained donor and the Pt(II)-contained acceptor in the necklace **1** generated the highest membrane-binding/damaging and bacterial killing ability (Figure R3, Supplementary Figure 22).

The preliminary results have revealed that the high antibacterial activity of necklace **1** is attributed to its more positive charges and the possible synergistic combination of heterometals in the necklace. Thus, the contribution of cell wall/plasma membrane disruption in bactericidal activity of the necklace **1** has been further confirmed by cell surface distribution of the necklace **1** and ultrathin section observations of the bacterial cells (Figure 5a, 5b). All results demonstrated that the heterometallic necklace **1** possessing more positive charges and two metal centers featured much stronger bacterium-binding activity and superior cell wall/plasma membrane-disrupting capacity than the control complexes, which might be mainly attributed to the synergistic combination of heterometals in the necklace (Figure 5c). From the above discussion, we may conclude that stoichiometric Cu(I)-contained donor and the Pt(II)-contained acceptor can be precisely integrated into the platform of necklace **1**, which is highly conducive to its outstanding bacterial killing ability.

We have added the corresponding description and related figures in both revised manuscript and Supporting Information:

“...An interesting observation is that all of the Cu(I)-containing molecules (*i.e.*, **1**, **2**, **D1**, and $\text{Cu}(\text{MeCN})_4\text{PF}_6$) had higher antibacterial activity than the Cu(I)-free

molecules (i.e., 3, 4, 5, 6, A1, and A2) (Figure 4d, Supplementary Figure 20, Table S1). In addition, we also found that all of the Cu(I)-containing molecules have remarkably stronger LPS-binding and membrane-damaging ability than the other molecules (Supplementary Figure 21)....”

“The contribution of cell wall/plasma membrane disruption in bactericidal activity of the necklace 1 was further confirmed by cell surface distribution of the necklace 1 and ultrathin section observations of the bacterial cells (Figure 5a, 5b). Energy dispersive spectroscopy (EDS) mapping showed that both Cu and Pt were abundantly accumulated on the cell surface and co-localized with the cellular elements (e.g., O, N, and P) (Figure 5a), indicating the direct contact between the necklace and the cell surface....”

Again, we greatly appreciate the reviewers' thoughtful suggestions that clearly improved the quality of our manuscript. With these changes and responses, we hope the revised manuscript is now acceptable for publication in *Nature Communications*.

With many thanks and best regard

Haibo

Prof. Dr. Hai-Bo Yang

Shanghai Key Laboratory of Green Chemistry and Chemical Processes

Department of Chemistry

East China Normal University

3663 N. Zhongshan Road

Shanghai, 200062, China

Phone:(+86)21-62235137

Cell: (+86)-15026691116

Reviewers' comments:

Reviewer #1 (Remarks to the Author):

The authors have addressed the comments, and I recommend acceptance of this manuscript.

Reviewer #2 (Remarks to the Author):

Referee report on the revised manuscript of NCOMMS-19-32134A

From my point of view I am satisfied with the changes made, concerning most of the identified issues and limitations.

However, there are still issues concerning the second crystal structure (225597_1_data_set_4391506_q52wr4.cif / hoa.res). Apparently, the checkcif file provided by the authors is not corresponding to the provided CIF. When running the checkcif routine directly on the CIF then additional A-alerts are present, resulting from missing values (e.g. for _diffrn_reflns_av_R_equivalents). I assume, that a wrong file has been uploaded. This has to be checked and resolved.

Additionally, the authors now included the following sentence into the manuscript: "The compound crystallizes in P-1 space group with the asymmetric unit is the whole triangular necklace 1 molecule. The large void inside the triangular necklace is filled with disordered solvents, which are removed using SQUEEZ routine of the Platon program during the crystal structure refinement."

Additionally to a typo (since it is "SQUEEZE" not "SQUEEZ") and improvable wording (since it is residual electron density, corresponding to disordered solvent molecules, which has been removed using the PLATON/SQUEEZE routine), actually the SQUEEZE routine is still not documented properly in the CIF. This is also the reason, why there exists the A-alert, warning for the large voids. Again, I strongly advise the authors to document that properly, since this can be resolved easily.

The sum formula is also wrong. This is stemming most likely from wrong AFIX cards for the calculated hydrogen atoms. For example there is an AFIX 93 for one of the methyl groups (COMA) of a P(Et3) ligand, which should be AFIX 137. This should always be carefully checked by the crystallographers, not only automatically calculated.

Also the argument by the authors, that the refinement takes too long is not valid! I tested a least-square refinement (4 cycles) on a normal tablet computer using 6 of the 8 processor nodes, and this took 132 seconds. See the attached result of SHELX-refinement:

```
+++++  
+ SHELXL - CRYSTAL STRUCTURE REFINEMENT - MULTI-CPU VERSION +  
+ Copyright(C) George M. Sheldrick 1993-2018 Version 2018/3 +
```

+ hoa started at 22:59:36 on 01-Mar-2020 +

+++++

Command line parameters: hoa -a50000 -b7371 -c624 -g0 -m0 -t6

-a sets the approximate maximum number of atoms including hydrogens.
-b sets the maximum number of full-matrix parameters (not used by CGLS).
For example -b9000 allows refinement of 1000 anisotropic atoms or 3000 with BLOC 1. For a 32-bit version, -b times the square root of the number of threads should not exceed about 65500. -c sets the reflection buffer size. This depends on the CPU cache size but will rarely need changing.

-g sets the number of reflection groups used for calculating R_complete, This must be greater than 1 but not greater than the total number of reflections for refinement. -m sets the current reflection group number. This may not be less than 1 nor greater than the number set by -g. These command line flags override other ways of defining free-R reflections. The -m value is also used as a seed for the WIGL pseudo-random shifts.

-t sets the number of threads, otherwise it is set to the apparent number of CPUs. For optimal performance on hyperthreading systems, -t should be set to a little more than half the number of CPUs; e.g. -t4 or -t5 for an Intel i7 processor.

Running 6 threads on 8 processors

Read instructions and data

** MERG code changed to 0 for compatibility with HKLF and BASF parameters **

wR2 = 0.3428 before cycle 1 for 60902 data and 2824 / 2824 parameters

GooF = S = 1.080; Restrained GooF = 1.081 for 213 restraints

Mean shift/esd = 0.000 Maximum = 0.010 for z Pt1 at 23:00:04

Max. shift = 0.000 A for H3NB Max. dU = 0.000 for C6BA

wR2 = 0.3428 before cycle 2 for 60902 data and 2824 / 2824 parameters

GooF = S = 1.080; Restrained GooF = 1.081 for 213 restraints

Mean shift/esd = 0.000 Maximum = 0.004 for z Pt1 at 23:00:34

Max. shift = 0.000 A for H3NB Max. dU = 0.000 for F7AA

wR2 = 0.3428 before cycle 3 for 60902 data and 2824 / 2824 parameters

GooF = S = 1.080; Restrained GooF = 1.081 for 213 restraints

Mean shift/esd = 0.000 Maximum = 0.002 for y C23^b at 23:01:09

Max. shift = 0.000 A for H3NB Max. dU = 0.000 for C6BA

wR2 = 0.3428 before cycle 4 for 60902 data and 2824 / 2824 parameters

GooF = S = 1.080; Restrained GooF = 1.081 for 213 restraints
Mean shift/esd = 0.000 Maximum = 0.002 for y C23^b at 23:01:43
Max. shift = 0.000 A for H3NB Max. dU = 0.000 for C6BA
wR2 = 0.3428 before cycle 5 for 60902 data and 0 / 2824 parameters
GooF = S = 1.080; Restrained GooF = 1.081 for 213 restraints
wR2 = 0.3428, GooF = S = 1.080, Restrained GooF = 1.081 for all data
R1 = 0.1052 for 53505 Fo > 4sig(Fo) and 0.1199 for all 60902 data
R.m.s. bond length deviation (from DFIX values) 0.0231 (A)
** Warning: 10 atoms may be split and 0 atoms NPD **
R1 = 0.1127 for 58779 unique reflections after merging for Fourier
Highest peak 5.99 at 0.3562 0.8560 0.4116 [0.13 A from PT6]
Deepest hole -1.75 at 0.0576 0.3141 0.1547 [0.76 A from PT2]

Please cite: G.M. Sheldrick (2015) "Crystal structure refinement with SHELXL", Acta Cryst., C71, 3-8 (Open Access) if SHELXL proves useful.

++++
+ hoa finished at 23:01:48 Total elapsed time: 132.56 secs +
++++

Of course, this is a bit longer than compared to small molecule structures, but the model can be definitely refined and documented properly within one day, even hours.

These severe (but mostly formal) issues certainly can and must be addressed, before the manuscript is suitable for publication. I am happy to check the revised file.

Reviewer #3 (Remarks to the Author):

The authors have made significant improvements. I would like recommend the publication of the manuscript in Nature Communications.

Response to Reviewer 2:

Reply: We greatly appreciate the reviewer's constructive suggestions on how to improve the quality of the crystallographic data in our manuscript. Accordingly, the further refinements have been carried out to address most of the issues, particularly A-alerts, solvent molecules and wrong formula problems that the reviewer concerned. The article of "Crystal structure refinement with SHELXL" was cited in the manuscript. The revised cif and checkcif files have been updated in the system.

Again, we greatly appreciate the reviewers' thoughtful suggestions that clearly improved the quality of our manuscript. With these changes and responses, we hope the revised manuscript is now acceptable for publication in *Nature Communications*.

With many thanks and best regard

Haibo

Prof. Dr. Hai-Bo Yang

Shanghai Key Laboratory of Green Chemistry and Chemical Processes

Department of Chemistry

East China Normal University

3663 N. Zhongshan Road

Shanghai, 200062, China

Phone:(+86)21-62235137

Cell: (+86)-15026691116

REVIEWERS' COMMENTS:

Reviewer #2 (Remarks to the Author):

I am satisfied with the changes made by the authors and can recommend the article for publication now.